# Redox Mechanisms Driving Skin Fibroblast-to-Myofibroblast Differentiation

**DOI:** 10.3390/antiox14040486

**Published:** 2025-04-18

**Authors:** Marzieh Aminzadehanboohi, Manousos Makridakis, Delphine Rasti, Yves Cambet, Karl-Heinz Krause, Antonia Vlahou, Vincent Jaquet

**Affiliations:** 1Department of Pathology and Immunology, Medical School, University of Geneva, 1211 Geneva, Switzerland; marzieh2424@gmail.com (M.A.); delphine.rasti@outlook.com (D.R.); karl-heinz.krause@unige.ch (K.-H.K.); 2Center of Systems Biology, Biomedical Research Foundation Academy of Athens, 11527 Athens, Greece; mmakrid@bioacademy.gr (M.M.); vlahoua@bioacademy.gr (A.V.); 3READS Unit, Medical School, University of Geneva, 1211 Geneva, Switzerland; yves.cambet@unige.ch

**Keywords:** proteomics, thiol oxidation, redox signalling, fibroblasts, myofibroblasts, transforming growth factor β1, epidermal growth factor

## Abstract

Transforming Growth Factor-Beta 1 (TGF-β1) plays a pivotal role in the differentiation of fibroblasts into myofibroblasts, which is a critical process in tissue repair, fibrosis, and wound healing. Upon exposure to TGF-β1, fibroblasts acquire a contractile phenotype and secrete collagen and extracellular matrix components. Numerous studies have identified hydrogen peroxide (H_2_O_2_) as a key downstream effector of TGF-β1 in this pathway. H_2_O_2_ functions as a signalling molecule, regulating various cellular processes mostly through post-translational redox modifications of cysteine thiol groups of specific proteins. In this study, we used primary human skin fibroblast cultures to investigate the oxidative mechanisms triggered by TGF-β1. We analyzed the expression of redox-related genes, evaluated the effects of the genetic and pharmacological inhibition of H_2_O_2_-producing enzymes, and employed an unbiased redox proteomics approach (OxICAT) to identify proteins undergoing reversible cysteine oxidation. Our findings revealed that TGF-β1 treatment upregulated the expression of oxidant-generating genes while downregulating antioxidant genes. Low concentrations of diphenyleneiodonium mitigated myofibroblast differentiation and mitochondrial oxygen consumption, suggesting the involvement of a flavoenzyme in this process. Furthermore, we identified the increased oxidation of highly conserved cysteine residues in key proteins such as the epidermal growth factor receptor, filamin A, fibulin-2, and endosialin during the differentiation process. Collectively, this study provides insights into the sources of H_2_O_2_ in fibroblasts and highlights the novel redox mechanisms underpinning fibroblast-to-myofibroblast differentiation.

## 1. Introduction

Fibroblasts are key cellular components of connective tissue, playing a pivotal role in maintaining tissue homeostasis, wound healing, and fibrosis. Fibroblasts are highly responsive to external stimuli, orchestrating dynamic cellular processes such as extracellular matrix (ECM) production, cytokine secretion, and differentiation into myofibroblasts [1]. Redox signalling is emerging as a central mediator of this process through a finely tuned mechanism involving reactive oxygen species (ROS) as signalling intermediates [2].

Redox signalling mostly operates through reversible post-translational oxidative modifications of specific amino acid residues, particularly cysteine thiols, in target proteins. These modifications influence protein function, localization, and interactions, enabling the precise regulation of cellular signalling pathways. Among ROS, hydrogen peroxide (H_2_O_2_) serves as a major secondary messenger in a reversible manner in a process reminiscent of phosphorylation [3]. The balance between ROS production and elimination is tightly regulated by enzymatic ROS generators, such as NADPH oxidases (NOXs), mitochondrial electron transport chains, as well as antioxidant systems, including catalase, glutathione peroxidases, peroxiredoxins, and thioredoxins.

Cysteine oxidation is a critical mechanism in redox signalling, where the reversible modification of cysteine thiol groups in proteins serves as a molecular switch to regulate various cellular processes [4]. ROS mediate the oxidation of cysteine residues, leading to the formation of sulfenic acid (–SOH), disulfides, or more stable modifications such as sulfinic (–SO_2_H) and sulfonic (–SO_3_H) acids. These modifications can alter protein function, localization, and interactions, enabling cells to respond dynamically to oxidative stress and signalling cues. Cysteine oxidation plays a pivotal role in maintaining cellular homeostasis and transmitting redox-based signals.

In fibroblasts, redox signalling has emerged as a key regulator of fibroblast phenotype and function, particularly during wound healing and pathological conditions such as fibrosis [5] and cancer [6]. The differentiation of fibroblasts into myofibroblasts, characterized by the increased expression of α-smooth muscle actin (α-SMA), leading to the enhanced contractility and production of ECM, is driven by pro-fibrotic cues like Transforming Growth Factor-Beta 1 (TGF-β1). Accumulating evidence indicates that ROS play a dual role in this process, acting as effectors of TGF-β1 signalling and inducing oxidative stress that exacerbates fibrosis. In particular, numerous studies have implicated H_2_O_2_ as a major downstream effector of TGF-β1-mediated fibroblast-to-myofibroblast differentiation [7,8].

It has been estimated that more than 40 enzymes can produce oxidants, such as H_2_O_2_ and the superoxide anion radical (O_2_**^·−^**), which are used as signalling agents [9]. Due to their potential as drug targets for fibrotic pathologies, there is a strong incentive to identify oxidant-producing enzymes governing fibroblast-to-myofibroblast differentiation. Among these, the H_2_O_2_-producing enzyme NOX4 has gained particular interest since it is massively upregulated in fibroblasts treated with TGF-β1 [10]. NOX4 inhibition has been shown to be anti-fibrotic in mouse models of the liver and pulmonary fibrosis [11,12] but inactive in kidney fibrosis, while the role of NOX4 has recently been challenged in TGF-β1-induced skin myofibroblast differentiation [13]. Furthermore, several other oxidant sources have been proposed to be involved in this mechanism, including DUOX1 [14], another member of the NADPH oxidase family, lysyl oxidases [15], and mitochondria [16].

Despite extensive research into NOX4 and other oxidant sources, the specific mechanisms by which ROS influence fibroblast biology remain incompletely understood. Key questions persist regarding the sources of ROS, the molecular targets of oxidative modifications, and the balance between physiological signalling and pathological oxidative damage. Addressing these questions is crucial for understanding the broader implications of redox biology in tissue repair and fibrosis, with potential therapeutic applications in diseases such as scleroderma, pulmonary fibrosis, and hypertrophic scarring.

The objective of this study was to identify oxidant sources activated during TGF-β1-mediated fibroblast differentiation, as well as downstream redox targets regulating this process. Redox proteomics is a powerful technique that allows the identification and quantification of redox-sensitive proteins, which undergo reversible oxidative modifications in response to changes in the cellular redox state [17]. By combining redox proteomics with biochemical and functional assays, we aimed to elucidate the molecular mechanisms underlying H_2_O_2_-mediated myofibroblast differentiation and identify potential targets for therapeutic intervention. We identified important changes in the expression of genes coding for redox proteins and identified several proteins as potential targets of redox signalling. Finally, our findings suggest a role for mitochondrial activity in fibroblast-to-myofibroblast differentiation.

## 2. Materials and Methods

### 2.1. Antibodies, Enzymes, and Reagents

Dulbecco’s Modified Eagle Medium (DMEM) containing 4.5 g/L glucose was purchased from Invitrogen. Fetal bovine serum (FBS) was obtained from Chemie Brunschwig (Basel, Switzerland) (K007310G). Phosphate-buffered saline (PBS; 14190) was purchased from Life Technologies (Paisley, Scotland, UK). TGF-β1 was purchased from BioLegend, (San Diego, CA, USA) and dissolved in 0.5% dimethyl sulfoxide (DMSO). Diphenylene iodonium chloride (DPI; CAS 4673-26-1) was obtained from Enzo Life Sciences (Lausen, Switzerland). GKT136901 (CAS 955272-06-7) was purchased from Alinda Chemical Ltd., Moscow, Russia. N-acetyl-L-cysteine (CAS 616-91-1), Trolox (CAS 53188-07-1), Edaravone (CAS 89-25-8), and GKT136901 (CAS 955272-06-7) were purchased from Sigma-Aldrich (Buchs, Switzerland).

### 2.2. Cell Culture

The human foreskin fibroblast cell line CCD-1112Sk (ATCC^®^ CRL-2429™) was maintained in DMEM supplemented with FBS (10%), penicillin (100 U/mL), and streptomycin (100 µg/mL) at 37 °C in the air with 5% CO_2_. HFF cells were passaged a maximum of 7 times. For differentiation experiments, the cells were kept in a serum-free medium for 24 h and treated with 2 ng/mL TGFβ1 with either DMSO or diphenylene iodonium (DPI) 1 µM for another 24 h. Cells were detached by the addition of trypsin/EDTA (0.05%), collected by centrifugation, and either passed or snap-frozen in liquid nitrogen and stored at −80 °C.

### 2.3. Gene Expression in Fibroblasts

RNA was extracted using the RNeasy Mini Kit (Qiagen, Dusseldorf, Germany) according to the manufacturer’s protocol. Concentration and purity (A260/A280 in the range of 1.9–2.1) were evaluated using a Nanodrop Spectrometer (Thermo Fisher Scientific). Five hundred nanograms (500 ng) of RNA were used for cDNA synthesis using the PrimeScript RT Reagent Kit (Takara, Saint-Germain-en-Laye, France) following the manufacturer’s instructions. Real-time PCR was performed using the SYBR Green (PowerUp SYBR Green Master Mix from Applied Biosystems, Thermo Fisher Scientific (Waltham, MA, USA) assay using 10 ng of cDNA and 0.3 µM of primers at the Genomics Platform, the National Center of Competence in Research-Frontiers in Genetics (Geneva, Switzerland), on a 7900HT SDS system (Applied Biosystems, Foster City, CA, USA). The efficiency of each primer was assessed with serial dilutions of cDNA. Relative expression levels were calculated by normalization to the geometric mean of two housekeeping genes, β2-microglobulin and GAPDH, as previously described [18]. Briefly, normalization was used as follows: relative quantities (RQs) were calculated with the formula RQ = E–Ct using efficiencies (E) calculated for each run with the DART-PCR algorithm, as described [19]. A mean quantity was calculated from triplicate PCR reactions for each sample, and this quantity was normalized to two or three similarly measured quantities of normalization genes as described [18]. Normalized quantities were averaged for three replicates for each data point and represented as the mean ± s.d. The highest normalized relative quantity was arbitrarily designated as a value of 1.0. Fold changes were calculated from the quotient of means of these normalized quantities and reported as ± s.d. The statistical significance of fold changes was determined by a paired Student’s *t*-test.

Normalized quantities are reported as the mean ± SEM. The sequences of the primers used in this study are documented in Appendix A.

### 2.4. Detection of H_2_O_2_ in Fibroblasts

Hydrogen peroxide (H_2_O_2_) production was quantified using the ROS-Glo™ H_2_O_2_ Assay (Promega), according to the manufacturer’s instructions. Fifty thousand (50,000) differentiated fibroblast cells were seeded per well in 96-well white plates with 100 μL of Hank’s Balanced Salt Solution (HBSS) per well. Six hours before the measurement, the culture medium was partially removed, leaving 80 μL per well, and 20 μL of the ROS-Glo™ H_2_O_2_ substrate solution, prepared using the manufacturer’s guidelines, was added to each well. The plates were incubated at 37 °C with 5% CO_2_. Twenty minutes prior to the measurement, 100 μL of the prepared ROS-Glo™ Detection Solution was added to each well. Luminescence was measured using a multi-mode microplate reader (FlexStation 3 or SpectraMax Paradigm, Molecular Devices, San Jose, CA, USA) with an emission wavelength of 450 nm.

### 2.5. Proteomic Analysis

#### 2.5.1. Sample Preparation for Total Protein Expression Analysis

Cell pellets were homogenized in a lysis buffer containing 8 M urea, 2% CHAPS, 50 mM Tris-HCl pH 6.8, 1 mM EDTA, and pH 7.0. Protein concentration was determined using the Bradford assay. Protease inhibitors (Roche) were added at a final concentration of 3.6%, and samples were stored at −80 °C until further use. Samples were processed with the GeLC-MS method as previously described [20]. Briefly, 10 μg of each sample was analyzed in SDS-PAGE. Electrophoresis was stopped when the samples just entered the separating gel. Gels were stained with Coomassie colloidal blue overnight. Each band was excised from the gel and further sliced into small pieces (1–2 mm). Gel pieces were destained with 40% acetonitrile, 50 mM NH_4_HCO_3_ and then reduced with 10 mM DTE in 100 mM NH_4_HCO_3_ for 20 min RT. After reduction, samples were alkylated with 54 mM Iodoacetamide in 100 mM NH_4_HCO_3_ for 20 min RT in the dark. Samples were then washed with 100 mM NH_4_HCO_3_ for 20 min, RT, followed by another wash with 40% acetonitrile, 50 mM NH_4_HCO_3_ for 20 min, RT, and a final wash with ultrapure water under the same conditions was performed. Gel pieces were dried in a centrifugal vacuum concentrator (speed vac) and trypsinized overnight in the dark, RT, by adding 600 ng trypsin per sample. Peptides were extracted after incubation with the following buffers: 50 mM NH_4_HCO_3_ for 15 min, RT, followed by two incubations with 10% formic acid, acetonitrile (1:1) for 15 min, RT. Peptides were eluted in a final volume of 600 μL and filtered with PVDF filters (Merck Millipore, Schaffausen, Switzerland) before being dried in a centrifugal vacuum concentrator (speed vac). Dried peptides were reconstituted in a mobile phase A buffer (0.1% formic acid, pH 3) and processed with LC-MS/MS analysis.

#### 2.5.2. LC-MS/MS Analysis for Total Protein Expression Analysis

In total, 2.5 μg of protein digest was loaded into a Dionex Ultimate 3000 RSLC nanoflow system (Dionex, Camberley, UK). After loading onto a Dionex 0.1 × 20 mm 5 μm C18 nanotrap column at a flow rate of 5 μL/min in 0.1% formic acid, samples were applied onto an Acclaim PepMap C18 nano-column (75 μm × 50 cm, 2 μm, 100 Å; Dionex, Sunnyvale, CA, USA), at a flow rate of 0.3 μL/min. The trap and nanoflow columns were maintained at 35 °C. The peptides were eluted with a gradient of solvent A: 0.1% formic acid versus solvent B: 80% acetonitrile and 0.1% formic acid for 480 min. The column was then washed and re-equilibrated prior to the injection of the next sample. The eluent was ionized using a Proxeon nanospray ESI source operating in the positive ion mode into an Orbitrap Velos FTMS (Thermo Finnigan, Bremen, Germany). The ionization voltage was 2.6 kV, and the capillary temperature was 275 °C. The mass-spectrometer was operated with MS/MS mode scanning from 380 to 1600 amu. The resolution of ions in MS1 was 60,000 and 7500 for HCD MS2. The top 20 multiply charged ions were selected from each scan for MS/MS analysis using HCD at 40% collision energy. Dynamic exclusion was enabled with a repeat count of 1 and an exclusion duration of 30 s.

#### 2.5.3. Data Processing for Total Protein Expression Analysis

Tandem mass spectra from the LC–MS/MS analysis of the samples were uploaded to Thermo Proteome Discoverer 1.4 software (Thermo Scientific, Hemel Hempstead, UK). Peptide and protein identification were performed with the SEQUEST search engine. A protein search was performed against the SwissProt human protein database downloaded on 28 January 2015, containing 20,197 reviewed entries. The following search parameters were applied: (i) precursor mass tolerance: 10 ppm and fragment mass tolerance: 0.05 Da; (ii) full tryptic digestion; (iii) maximum missed cleavage sites: 2; (iv) static modifications: carbamidomethylation of cysteine; (v) dynamic modifications: oxidation of methionine; (vi) target FDR (strict): 0.01; and (vii) target FDR (relaxed): 0.05. The identified peptides were exported from “Proteome Discoverer”, and label-free quantification was performed as described previously [21]. Peptides consistently reported in more than 60% of the samples (at least in one group) were included in the differential expression analysis. Statistical analysis was performed using the Mann–Whitney test. Proteins with *p*-values ≤0.05 and ratios of ≥1.5 or ≤0.66 were considered statistically significant.

#### 2.5.4. Pathway Enrichment Analysis

Pathway enrichment analysis was performed using Cytoscape’s (http://www.genome.org/cgi/doi/10.1101/gr.1239303), accessed on 1 April 2016, version 3.3.0 ClueGO (https://doi.org/10.1093/bioinformatics/btp101) plug-in accessed on 1 April 2016. Briefly, the REACTOME pathway database was used, and only statistically significant pathways (Bonferroni correction *p*-value ≤ 0.05, two-sided hypergeometric test) were considered for the analysis. The remaining settings were used as the default.

### 2.6. OxICAT Analysis

#### 2.6.1. Sample Preparation for Cysteine Oxidation Analysis Using ICAT (OxICAT)

The cell lysates were dissolved in 8 M urea, 2% CHAPS, 50 mM Tris-HCl pH 6.8, 1 mM EDTA, and pH 7.0. A volume corresponding to 200 μg per lysate was mixed with the appropriate volume of DAB buffer (Denaturing Alkylation Buffer) consisting of 6 M urea, 0.5% (*w*/*v*) SDS, 10 mM EDTA, 200 mM Tris_HCl, and pH 8.5 so that a final volume of 100 μL per lysate was achieved. Lysates were subjected to buffer exchange with 7 kDa MWCO Zeba Spin filters (Thermo Scientific). After buffer exchange, the lysates were eluted in 80 μL of the DAB buffer. A cleavable light ICAT reagent (AB Sciex) was dissolved in 20 μL of acetonitrile (ACN). Each lysate (eluted in 80 μL DAB buffer) was mixed with the dissolved content of one vial of the cleavable light ICAT reagent (20 μL). The mixture of the lysate-light ICAT reagent was incubated for 2 h at 37 °C in the dark (to label reduced-SH thiols). The excess of the light ICAT reagent was removed with 7 kDa MWCO Zeba Spin filters, and light ICAT-labelled lysates were eluted in an 80 μL DAB buffer. The reduction of reversiblly oxidized thiols (SOH, SNO, SS, SO_2_H) was performed by incubating the lysate with 1.25 mM TCEP [Tris(2-carboxyethyl)phosphine hydrochloride] for 30 min at 37 °C. The cleavable heavy ICAT reagent (AB Sciex) was dissolved in 20 μL of ACN. The reduced sample (80 μL) was mixed with the dissolved content of one vial of the cleavable-heavy ICAT reagent (20 μL) and incubated for 2 h at 37 °C in the dark (labelling of reversible oxidized thiols: SOH, SNO, SS, and SO_2_H). The excess of the heavy ICAT reagent was removed with 7 kDa MWCO Zeba Spin filters. The same filters were also used to perform buffer exchange, and the lysates were eluted in 80 μL of 0.1% SDS, 50 mM Tris-HCl, and pH 8.5. A final lysate volume of 100 μL prior to trypsinization was achieved with the addition of 20 μL ACN to each lysate. Trypsinization was performed overnight in the dark, RT (trypsin/protein = 1:16), according to the manufacturer’s instructions. After trypsinization, the lysates were desalted with strong cation exchange chromatography (SCX cartridge), followed by cysteine-containing peptide enrichment via avidin chromatography (light and heavy ICAT reagents were biotinylated). The final peptide elution of avidin chromatography was dried in a centrifugal vacuum concentrator (speed vac). The peptides were subjected to biotin cleavage and dried again in a speed vac. Dried peptides were stored at −80 °C until use [22,23].

#### 2.6.2. OxICAT LC-MS/MS Analysis

The LC-MS/MS analysis of OxICAT samples was performed as described above using the same Dionex system. However, samples were eluted with a 120 min gradient of solvent A (0.1% formic acid) versus solvent B (80% acetonitrile, 0.1% formic acid). The eluent was ionized using a Proxeon nanospray ESI source operating in the positive ion mode into an Orbitrap Velos FTMS with the same instrument settings as described above.

#### 2.6.3. Data Processing for OxICAT Analysis

Tandem mass spectra from the LC–MS/MS analysis of the OxICAT samples were uploaded to Proteome Discoverer v1.4 software. Peptide and protein identification were performed as described above with the same SwissProt human protein database. The following search parameters were applied: (i) precursor mass tolerance: 10 ppm and fragment mass tolerance: 0.05 Da; (ii) full tryptic digestion; (iii) max missed cleavage sites: 2; (iv) dynamic modifications: the oxidation of methionine, deamidation of asparagine and glutamate, acetylation (any N terminus), and ICAT-C, ICAT-C:13C(9); and (v) target FDR (strict): 0.01 and (x) target FDR (relaxed): 0.05. The obtained results were further processed by applying the following filters: (i) high and medium confidence peptides (FDR < 5%) and (ii) peptide rank 1. Subsequently, label-based quantification was performed at the peptide level based on the ICAT reporter ion intensities detected by the Reporter Ions Quantifier Node in Proteome Discoverer. For each distinct peptide, the abundance was calculated as the mean of reported ions from all matching spectra, with spectra grouped by sequence. The reporter ion intensities for each individual peptide were represented as the ratio of heavy-labelled to light-labelled peptides. Only peptides for which both heavy and light reporter ion-labelled peptides were detected and quantified were considered for further analysis. Cysteine residues where only light or heavy reporter ions were detected, probably due to the low signal intensity of the other respective labelled peptides, were discarded to minimize distortions due to peptide abundance. The mean heavy-to-light (H/L) ratio for each labelled peptide per group was calculated. We only considered cysteine residues that were labelled with both light and heavy ICAT labels in at least two biological replicates out of five per group. These calculations were performed with Excel spreadsheet calculation software (Microsoft, Redmond, WA, USA). Subsequently, to determine the fold change in the oxidation status of the peptide in a specific comparison, the ratio of the respective H/L ratios was calculated. Labelled peptides with fold changes ≥1.2 or ≤0.8 were considered to show a significant change in their oxidation status. Moreover, to assess the consistency of those shifts in fold change oxidation status and increase the validity of the observed changes, all individual ratios of the H/L ratio of each sample per group were calculated in each specific comparison. When more than 60% of the observed individual ratios were also ≥1.2 or ≤0.8, respectively, in agreement with the mean fold change, these changes were considered significant.

### 2.7. EGFR Dimerization Analysis

#### 2.7.1. Cross-Linking of EGFR Dimer in Cultured Fibroblasts

HFF cells were cultured as described above. EGFR dimerization was detected following cross-linking with bis[sulfosuccinimidyl] suberate (BS^3^), as described [24], with minor modifications. After 23 h of vehicle, TGFβ1 and DPI treatment, the cells were placed on ice for 1 h in order to prevent endocytosis of the EGFR dimer. For the last 2 min of this hour, cells were supplemented with 50 ng/mL of the EGF or vehicle. All procedures from this point on were performed on ice. After aspiration of the medium, the cells were washed three times with 1× ice-cold PBS. The cells were treated with 3 mM BS^3^ in 1× ice-cold PBS for 20 min, and then excess BS^3^ was quenched with double the volume of 250 mM glycine in 1× ice-cold PBS for 5 min. After the aspiration of the liquid, the cells were washed three times with ice-cold 1× PBS and fresh PBS was added to the flasks. The cell layer of each flask was scraped with a cell scraper, and the cells were moved to a clean falcon tube. Cells were harvested via centrifugation at 1500× *g* for 5 min at 4 °C, and the supernatant was discarded. The pellet was gently dissolved in a homogenization buffer [50 mM Tris-HCl pH 8.8; 250 mM sucrose; 2 mM EDTA; 1 mM EGTA; 50 mM sodium fluoride; 1% Triton X-100; 100 μM activated sodium orthovanadate; 10 mM β-mercaptoethanol; 3.6% protease inhibitor cocktail (Roche)] and transferred to a clean Eppendorf tube. The suspension was then passed once through a 29 G needle as hard as possible, and the flushed suspension was left on ice for 30 min. The suspension was then centrifuged at 16,000× *g* for 20 min at 4 °C, and the supernatant was transferred to a clean Eppendorf tube. The lysate was stored at −20 °C for Western blot experiments.

#### 2.7.2. Western Blot

EGFR detection was performed using Western blot analysis. In brief, for the Western blot, 25 μL of cell lysate from each treatment was mixed with Laemmli’s buffer and loaded in 6% SDS polyacrylamide gel after incubation at 90 °C for 10 min. In parallel, another 6% SDS polyacrylamide gel was run at the same conditions as a protein loading control with 10 μL of the cell lysate from each treatment and was then stained overnight with Coomassie Colloidal Blue. The gels were run at 40 V for 15 min and then at 80 V for ~2.5 h until the 75 kDa band of the molecular weight marker approached the bottom of the gel. Proteins were then transferred to a nitrocellulose membrane at 290 mA for 2 h at 4 °C. After blocking for 2 h with bovine serum albumin (BSA) in TBS-Tween 0.1% *v*/*v* at room temperature, the membrane was washed three times with TBS-Tween 0.1% *v*/*v* and incubated overnight with the primary antibody at 4 °C. All antibodies were diluted in a blocking buffer according to the manufacturers’ recommendations: anti-EGFR [Cell Signalling Technologies, EGF Receptor (D38B1), #4267]: 1:1000; secondary anti-rabbit [Amersham Biosciences, Anti-rabbit IgG, peroxidase-linked species-specific F(ab′)2 fragment (from donkey), NA 9340]: 1:10,000. The membrane was washed again three times with TBS-Tween 0.1% *v*/*v* and incubated for 2 h with the secondary antibody at room temperature. The membrane was then washed three times with TBS-Tween 0.1% *v*/*v* and incubated with ECL for 5 min in the dark. Film exposure and development followed. Films and gels were scanned using a GS-800 imaging densitometer (Biorad) in the transmission mode. The optical density of the bands on the scanned films and corresponding gels was measured using Quantity One 1-D Analysis Software (Biorad). Statistical analysis was performed using the Mann–Whitney U test in IBM SPSS Statistics v.22 software.

### 2.8. Gene Silencing in HFF (siRNA)

The human foreskin fibroblast cell line was maintained in DMEM supplemented with FBS (10%), penicillin (100 U/mL), and streptomycin (100 µg/mL) at 37 °C in air with 5% CO_2_ in a 6-well plate for 24 h. The next day, the DMEM was replaced with 1750 µL of fresh serum-free DMEM, and 2 ng/mL of TGFβ was added to the relevant conditions. A transfection mixture was prepared for each well in duplicate, consisting of 7.5 µL of the TransIT-X2 reagent and 6.8 µL of siRNA, with a final concentration of 25 nM for each siRNA (Thermofisher s224161 for NOX4, s28799 for DUOX1, s8262 for LOXL2, and s10838 for POR), in 250 µL of serum-free medium. Following transfection, the cells were incubated at 37 °C with 5% CO_2_ for 24 h. The incubation ended by removing the medium, washing with PBS 1X, and directly adding the RLT solution from the lysing of cells using the RLT solution of RNAeasy Mini Kit from Qiagen.

### 2.9. Oxygen Consumption Rate (OCR)

OCR was performed using the XF Cell Mito Stress Test Kit according to the protocol. XF Analyzer was warmed up overnight. Fibroblasts (25,000 cells in 200 µL of FBS-free DMEM), with and without 2 ng/mL TGFβ, were seeded into the Seahorse 96-well plate and incubated overnight. The sensor cartridge was hydrated in the XF Calibrant at 37 °C in a non-CO_2_ incubator overnight. Stock Compounds (oligomycin, FCCP, and rotenone) were prepared by incubating the compound for 15 min at room temperature. Following that, the compounds were resuspended, resulting in final concentrations of 100 µM oligomycin, 100 µM FCCP, and 50 µM rotenone. Cells were washed, and the cell culture medium was replaced with a pre-warmed assay medium supplemented with 1 mM pyruvate, 2 mM glutamine, and 10 mM glucose (pH adjusted to 7.4), followed by incubation at 37 °C without CO_2_ for 45 min to 1 hour. After calibrating the sensor cartridge for 15–30 min, the cell culture microplate was placed on the instrument tray. After the assay run was completed, we read the OCR values measured by the XF Analyzer.

### 2.10. Sequence Comparison

The accession numbers used for the identification of cysteine conservation are documented in Appendix A.

## 3. Results

### 3.1. Changes in Expression Redox Genes in Fibroblasts Following TGFβ1 Treatment

We first addressed the impact of TGF-β1 treatment on the expression of a selection of pro- and antioxidant genes in fibroblasts using qPCR. We used the upregulation of the myofibroblast markers α-SMA, fibronectin, and peroxidasin (ACTA2, FN1, and PXDN) as confirmation of successful myofibroblast differentiation (Figure 1A). The selected genes included NADPH oxidases (NOX) 1–5, dual oxidases (DUOX) 1 and 2, peroxiredoxins (PRDX) 1–6, glutathione peroxidases (GPX) 1–8, glutaredoxins (GLRX) 1–5, thioredoxins (TXN, TXNRD 1–3, TXN2, TXNDC5, TXNIP), and genes encoding redox proteins of the endoplasmic reticulum (ERO1a, ERO1b, QSOX1, QSOX2, SELENOF, and VKORC1) (Figure 1B). Several peroxiredoxin isoforms (PRDX6, 3, 4, 1, and 5), key regulators of intracellular H_2_O_2_, were upregulated, suggesting a possible increase in intracellular redox signalling. The expression of enzymes involved in neutralizing H_2_O_2_, such as catalase and glutathione peroxidases 1 and 4, was significantly decreased. NOX isoforms were either not expressed (NOX2, NOX3), unchanged (DUOX1, DUOX2), slightly upregulated (NOX1), or downregulated (NOX5). However, the H_2_O_2_-producing enzyme NOX4 expression was highly upregulated, suggesting its potential role in this process (Figure 1C).

### 3.2. TGFβ1-Mediated Myofibroblast Differentiation Is Mitigated by Antioxidant Compounds

We tested whether pharmacological compounds known to interfere with oxidant production affect TGFβ1-induced myofibroblast differentiation. To test this, we treated the fibroblasts with TGFβ1 and the flavoprotein inhibitor DPI, the glutathione precursor N-acetylcysteine, the water-soluble analogue of vitamin E, Trolox, the free radical scavenger Edaravone, and GKT136901, which is an antioxidant often used as a NOX1/4 inhibitor. After 24 h, we extracted mRNA and performed qPCR, finding that all antioxidant compounds were able to reduce ACTA2 expression, confirming a global redox mechanism in this process (Figure 1D). To assess the functional impact of redox gene expression changes on human skin fibroblasts, we measured global H_2_O_2_ production following TGFβ1 stimulation using the bioluminescent ROS-Glo™ assay. We showed that H_2_O_2_ levels were significantly increased following TGFβ1 treatment. Fibroblasts treated with TGFβ1 exhibited an increased bioluminescent signal compared to the vehicle control. The signal was inhibited by catalase, proving the specificity to be H_2_O_2_, while inhibition of the signal by diphenylene iodonium (DPI) indicated that the source of H_2_O_2_ is most likely a flavoprotein (Figure 1E). We considered that the high potency of DPI was due to the specific inhibition of an oxidant-generating flavoenzyme and performed the following experiments using this compound. 

### 3.3. Total Protein Expression Analysis of Fibroblasts Shows That Upregulation of Collagen Synthesis in TGFβ1-Treated Fibroblasts Is Mitigated by DPI

Based on the above, we performed a proteomic analysis. Fibroblasts treated with TGFβ1, a vehicle (DMSO), and TGFβ1/DPI were analyzed for their total protein expression profile by LC-MS/MS followed by label-free protein quantification (five biological replicates per group), and differentially expressed proteins were determined [proteins with a *p*-value ≤ 0.05 (Mann–Whitney test) and a ratio ≥1.5 or ≤0.66]. Several members of the collagen family and proteins involved in ECM-associated proteins were upregulated in the presence of TGFβ1 (Figure 2A and Appendix A), which are all hallmarks of myofibroblast differentiation. DPI significantly mitigated the upregulation of proteins related to collagen formation and ECM organization in TGFβ1-treated fibroblasts (Figure 2B and Appendix A) but also induced important changes in normal fibroblasts (Figure 2C and Appendix A).

The pathway mapping of the differentially expressed proteins was performed using Cytoscape open-access software (Appendix A). Proteins involved in translation and protein folding were downregulated by TGF-β1. In addition, interestingly, glucose-6-phosphate dehydrogenase (G6PD), the rate-limiting enzyme in the pentose phosphate pathway (PPP), as well as Transketolase (TKT), another PPP enzyme, were significantly down-regulated, supporting the fact that PPP was affected in TGFβ1-treated fibroblasts. DPI primarily mitigated collagen synthesis (Figure 2D) but did not affect several other molecular pathways induced by TGFβ1, such as PPP (Figure 2B,C), suggesting that other non-redox regulatory mechanisms are also involved.

### 3.4. OxICAT Analysis Detected Cysteine Oxidation Changes After TGFβ1 Treatment That Can Be Reversed by DPI

The proteomic analysis of cysteine-oxidation levels was performed using the OxICAT methodology in fibroblasts treated with TGF-β1, a vehicle and TGFβ1/DPI for 24 h (Figure 3). The OxICAT method involves the enrichment of cysteine-containing peptides, the differential labelling of reduced (light ICAT label) and reversibly oxidized (heavy ICAT label) cysteines per cell, the estimation of the oxidized-to-reduced cysteine ratio (H/L) per cell and a comparison of these ratios between the different cell treatments by calculating the respective fold change reflecting differences in the oxidation status of each cysteine-containing peptide (Figure 3). Data analysis was conducted as described previously [25]. For a peptide to be considered redox-sensitive, several requirements had to be fulfilled to increase the reliability of analysis and highlight the most biologically relevant oxidation events: cysteine residues had to be detected at both reversibly reduced and oxidized states (e.g., a ratio of H/L can be calculated) in at least two biological replicates per group. A comparison of these ratios between two cell states should indicate a ≥1.2-fold change; the observed shift in ≥1.2 had to be observed in at least 60% of the observed individual fold changes to increase the validity of the observed changes. Special emphasis was placed on the oxidation events induced by TGF-β1 that were reversed following DPI treatment.

Supporting the redox impact of the TGFβ1 response, several proteins met all selection criteria (Appendix A). Among these, filamin A, an actin-binding protein; fibulin-2, an ECM component; endosialin, a receptor involved in cell adhesion; and the epidermal growth factor receptor (EGFR), were selected as more prominent findings were apparently differentially oxidized following TGF treatment (Figure 4A). Supporting this observation, the proteomic analysis of fibroblasts indicated that the levels of expression of these proteins were unchanged in the tested conditions, suggesting that the observed changes in oxidation status were not due to differential protein expression (Appendix A).

Importantly, the conservation of cysteine residues across species often indicates functional relevance. We performed alignments of protein sequences for the cysteines identified by OxICAT using available sequences from human (*H. sapiens*), mouse (*M. musculus*), chicken (*G. gallus*), frog (*X. tropicalis*), and zebrafish (*D. rerio*). Although neighbouring residues are poorly conserved, the alignment revealed a high conservation of all identified cysteines across the species, suggesting that these residues are evolutionarily preserved to maintain specific protein functions (Appendix A).

### 3.5. EGFR Dimers Are Stabilized in TGFβ1-Treated Fibroblasts

Ligand binding to the EGFR leads to dimerization, trans-autophosphorylation, and activation [26]. Cys307, which was found to be oxidized after TGF-β1 treatment, is located in extracellular domain II of the receptor—which is a region critical for dimerization [27]. Dimerization is essential for the intracellular kinase domain of EGFR to become activated. To investigate the impact of TGF-β1 on the stability of the EGFR dimer, fibroblasts were treated for 24 h with TGF-β1, followed by a 5 min EGF stimulation to induce dimerization. The EGFR dimer was crosslinked using bis[sulfosuccinimidyl] suberate (BS^3^) and detected by Western blot (Figure 4B,C). The treatment of fibroblasts with TGF-β1 resulted in significantly higher levels of EGFR dimerization, which were diminished upon DPI addition, confirming the OxICAT findings.

### 3.6. Potential Enzymatic ROS Sources Do Not Impact Myofibroblast Marker Expression Following TGFβ1

To address the potential sources of reactive oxygen species (ROS) that might influence fibroblast differentiation into myofibroblasts, we used siRNA to silence several candidate ROS-generating genes, including NOX4, LOXL2, POR, and DUOX1, following treatment with TGF-β1. We measured the expression of key myofibroblast markers, α-smooth muscle actin (α-SMA, encoded by ACTA2), and fibronectin (encoded by FN1). Knocking down any of these genes did not reduce the marker expression. In contrast, treatment with diphenyleneiodonium (DPI), a general ROS inhibitor, significantly lowered the levels of both ACTA2 (the gene-encoding α-SMA) and FN1 (the gene-encoding fibronectin) (Figure 5A). Confirming siRNA efficacy, we detected a clear decrease in the expression of the targeted genes without affecting other non-targeted genes (Figure 5B).

### 3.7. TGFβ1 Promotes Metabolic Reprogramming in Fibroblasts

We further investigated the metabolic effects of TGFβ1 on HFF cells by performing the oxygen consumption rate (OCR) and extracellular acidification rate (ECAR) analyses of fibroblasts treated with TGFβ1 and TGFβ1/DPI. OCR measurements revealed that basal respiration, ATP-dependent respiration, and maximal respiration were all significantly increased in response to TGFβ1. However, co-treatment with DPI markedly decreased these parameters (Figure 6A). ECAR analysis also demonstrated that TGF-β1 enhanced basal glycolysis, glycolytic capacity, and glycolytic acidification, which is consistent with an increase in glycolytic activity. Consistent with the inhibition of mitochondrial respiration, DPI treatment induced a striking metabolic shift towards glycolysis (Figure 6B). Together, these findings reveal that TGF-β1 drives the metabolic reprogramming of fibroblasts by enhancing mitochondrial respiration and glycolysis, both of which are predominantly dependent on ROS signalling.

## 4. Discussion

The differentiation of skin fibroblasts into myofibroblasts by TGF-β1 is a critical process in wound healing and fibrosis [1,28,29]. Our study explored the role of redox mechanisms in regulating this transition, demonstrating that oxidative processes—primarily mediated by H_2_O_2_—act as key drivers of fibroblast-to-myofibroblast differentiation by modulating signalling pathways, inducing metabolic changes and specific redox cysteine modifications.

The treatment of human skin fibroblasts in vitro with TGF-β1 induced H_2_O_2_ generation and a global decrease in antioxidant enzymes, particularly catalase and glutathione peroxidases, which play a crucial role in H_2_O_2_ catabolism [30]. Additionally, TGF-β1 treatment led to a significant upregulation of NOX4, which is an enzyme dedicated to H_2_O_2_ production [31]. Increased NOX4 expression and H_2_O_2_ levels in fibroblasts treated with TGF-β1 have previously been reported in numerous fibroblast types, where NOX4 has been proposed as a key regulator of fibroblast-to-myofibroblast differentiation [5,32]. However, using siRNA, our study indicated that NOX4 is not involved in this process, reinforcing previous findings from our laboratory [13].

The inhibition of fibroblast-to-myofibroblast differentiation, as indicated by decreased αSMA expression following TGF-β1 treatment using antioxidant molecules and DPI, suggests that alternative enzymatic sources of H_2_O_2_ contribute to this process. However, none of the tested H_2_O_2_-generating enzymes—DUOX1 [14], LOXL2 [33], or POR (a crucial enzyme for CYP450 synthesis) [34]—were found to be involved. While we cannot rule out the possibility of a coordinated contribution from multiple enzymatic systems or other oxidant sources, our data point toward alternative sources of ROS and possibly a mitochondrial role in this process.

TGF-β1-treated fibroblasts exhibited increased basal respiration compared to non-treated fibroblasts, which was strongly inhibited by low concentrations of DPI. Although DPI is commonly described as a NOX inhibitor, and its ability to mitigate myofibroblast differentiation has been attributed to NOX4 inhibition [5,10], its profound impact on mitochondrial respiration has often been overlooked. Notably, DPI is also a potent inhibitor of mitochondrial complex I [35], leading to the blockade of the mitochondrial electron transport chain and triggering a significant metabolic shift in fibroblasts. Both TGFβ1 and DPI enhanced glycolysis. However, DPI simultaneously impaired mitochondrial function and increased ROS production, creating a metabolic environment that may be incompatible with myofibroblast differentiation and highlighting the fact that glycolysis alone may not be sufficient to drive fibroblast-to-myofibroblast differentiation.

DPI not only effectively suppressed TGF-β1-induced fibroblast-to-myofibroblast differentiation but also strongly inhibited mitochondrial respiration. While mitochondrial ROS are traditionally considered mere by-products of oxidative phosphorylation and are often linked to cellular dysfunction and disease, emerging evidence highlights their crucial signalling functions. In particular, mitochondrial H_2_O_2_ has been identified as a key redox mediator regulating various cellular processes, including adaptation and stress resistance [36,37].

Several studies have demonstrated the involvement of mitochondrial oxidants in fibroblast differentiation [16,38], including an increase in the mitochondrial number and activity during TGF-β1-induced differentiation. These findings suggest that fibroblast-to-myofibroblast transition is accompanied by significant metabolic remodelling. Along this line, in addition to the increase in typical markers of myofibroblasts, such as COL1A1, COL1A2 and COL2A1, we identified several modified metabolic pathways in the proteomic analysis of TGF-β1-treated fibroblasts. While our findings suggest the possible role of mitochondrial redox signalling in fibroblast differentiation, further studies using specific mitochondrial-targeted probes, such as MitoQ and MitoSOX, are needed to directly confirm this involvement.

The OxICAT approach identifies the ratio of oxidized to reduced forms of cysteines, providing insights into potential downstream targets of oxidant production occurring in fibroblasts following TGF-β1 treatment. This is, to our knowledge, the first study to address redox modifications in TGFβ1-induced fibroblast-to-myofibroblast differentiation. However, a study by Katrin Schröder’s group used the BIAM-based redox switch assay in combination with mass spectrometry and Western blot analysis to analyze differential oxidation in podocytes isolated from WT and Nox4-/- mice treated with TGFβ1 [39]. In their study, it was identified that Nox4 is induced by TGFβ1 and leads to the oxidation of numerous downstream targets. In particular, a significant enrichment of peroxiredoxins and thioredoxins was found, which are key redox relays in redox signalling [40]. In our study, we surprisingly did not find such an enrichment of peroxiredoxins and thioredoxins. This discrepancy may be due to significant differences in the whole experimental pipelines of the studies, including the cell context (podocytes versus fibroblasts) but also the type of cysteine oxidation analysis used. In our study, we calculated the ratio of reversible oxidized versus reduced cysteine per sample, whereas the information on this ratio was not considered in the study by Schröder.

We focused on peptides, which showed increased oxidation, following TGF-β1 treatment, which was decreased by DPI, suggesting its potential role in fibroblast-to-myofibroblast differentiation. The included peptides contained specific cysteine residues of EGFR, filamin A, fibulin 2, and endosialin.

While the involvement of filamin A, fibulin 2, and endosialin in various cellular processes is well-established, specific studies directly linking these proteins to fibroblast-to-myofibroblast differentiation are limited, but the high conservation of redox-sensitive cysteines during evolution indicates potential important roles, which awaits further confirmation.

However, EGFR is known to be regulated by redox mechanisms [41] and is induced by TGF-β1 during fibroblast-to-myofibroblast differentiation [42]. In particular, Cys797, located in the EGFR kinase domain, is crucial for receptor activation and is subject to redox regulation through reversible oxidation, which can modulate its catalytic activity [43]. The oxidation of Cys797 can inhibit the EGFR function by preventing ATP binding, while its reduction restores activity, highlighting its role as a redox-sensitive switch in EGFR signalling [41]. The redox-sensitive nature of Cys797 is exploited for the pharmacological inhibition of EGFR by covalent tyrosine kinase inhibitors (e.g., osimertinib), which form irreversible bonds with the thiol group of Cys797, preventing ATP binding and inhibiting EGFR activity [44].

We did not detect oxidation of Cys797 but identified that the Cys307 oxidation of EGFR was increased following TGF-β1 and mitigated following DPI treatment. The Cys307 of EGFR is located in extracellular domain II, a critical region for receptor dimerization [26]. Notably, Cys307 was found to be oxidized following TGF-β1 treatment, coinciding with increased EGFR dimerization. This effect was mitigated by DPI treatment, suggesting the role of oxidative modifications in EGFR activation. While specific studies on Cys307 are limited, Cys307 has been shown to undergo reversible oxidation in HeLa cells following 5 min of EGF stimulation, further supporting its role in EGFR activation [45].

Interestingly, a common EGFR mutation found in glioblastoma—the de2-7EGFR truncation (EGFRvIII)—disrupts Cys295-Cys307 cysteine pairing due to the absence of Cys295. This alteration leaves Cys307 unpaired, resulting in a constitutively active receptor that drives uncontrolled cell signalling and proliferation [46]. This suggests that Cys307 plays a crucial role in EGFR activation and dimerization through redox regulation, and its disruption in the EGFRvIII mutant may contribute to oncogenic signalling, highlighting its potential as a therapeutic target in redox-driven EGFR activation and cancer progression.

Despite these insights, some limitations of our study must be acknowledged. This study is primarily descriptive and does not explore the functional consequences of cysteine redox modulation. In particular, EGFR oxidation following TGF-β1 stimulation may result from changes in its localization, exposing it to different redox environments—an oxidative extracellular space or a reducing intracellular compartment during internalization. The in vitro nature of our experiments, while informative, does not fully replicate the complex microenvironment of wound healing and fibrosis in vivo. Future studies utilizing animal models or human tissue samples will be necessary to validate our findings and assess the translational potential of redox-targeted therapies.

## 5. Conclusions

Our findings highlight the critical role of redox regulation in fibroblast-to-myofibroblast differentiation, emphasizing H_2_O_2_-mediated signalling and mitochondrial involvement in this process. While previous studies have implicated NOX4 as a key regulator, our data suggest that alternative enzymatic or mitochondrial sources drive redox signalling during TGF-β1-induced differentiation. Through OxICAT proteomic analysis, we identified key cysteine residues—particularly in EGFR, filamin A, fibulin 2, and endosialin—that undergo redox modifications, potentially linking these proteins to fibroblast activation. Among them, EGFR emerged as a central redox-regulated target, with the oxidation of Cys307 playing a crucial role in receptor dimerization. These findings open avenues for developing redox-based therapeutic strategies to regulate fibrosis and improve wound healing outcomes.

## Figures and Tables

**Figure 1 antioxidants-14-00486-f001:**
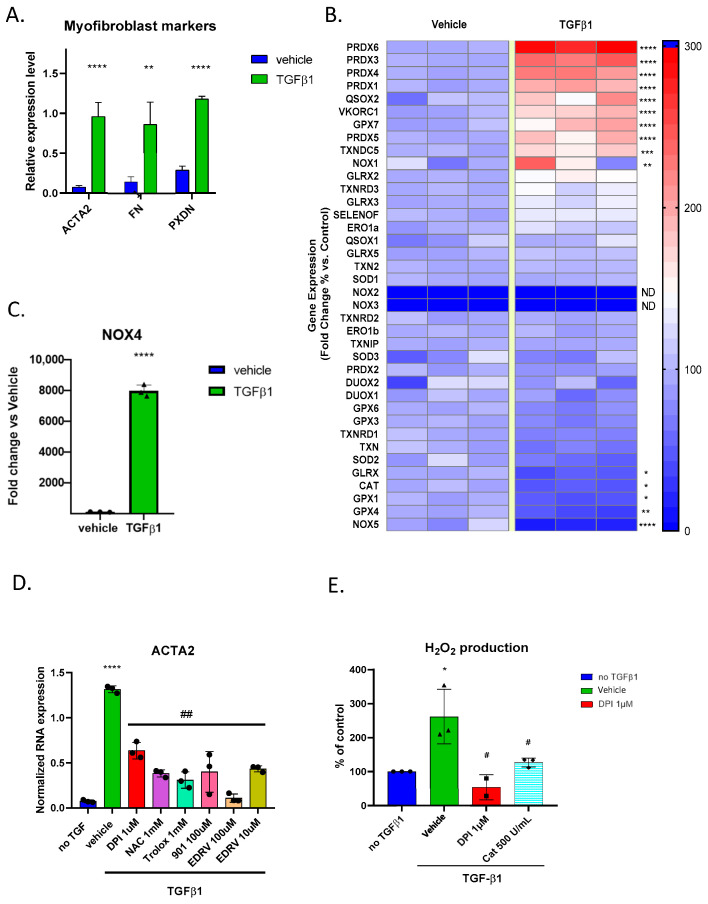
Redox gene expression and oxidative stress response in fibroblasts treated with TGF-β1: (**A**) Bar graph showing upregulation in typical myofibroblast markers ACTA2, FN1, and PXDN following the TGFβ1 treatment of human skin fibroblasts; (**B**) heatmap visualization of RT-qPCR analysis (% of fold change) of redox-related genes in human foreskin fibroblasts before and after TGF-β1 stimulation. Red represents up-regulation, and blue represents down-regulation; (**C**) bar graph showing a massive increase in NOX4 mRNA expression in fibroblasts treated with TGF-β1 compared to vehicle control, expressed as % of fold change; (**D**) bar graph showing that the upregulation of the myofibroblast marker ACTA2 is mitigated by 24 h of treatment of several antioxidant molecules. Error bars represent the standard deviation of biological replicates. (**E**) Bar graph showing increased H_2_O_2_ levels in fibroblasts treated with TGF-β1 and mitigation by DPI and catalase. Statistical test and two-tailed unpaired *t*-test. * *p* < 0.05, ** *p* < 0.01, *** *p* < 0.005, **** *p* < 0.001, # *p* < 0.05, ## *p* < 0.01.

**Figure 2 antioxidants-14-00486-f002:**
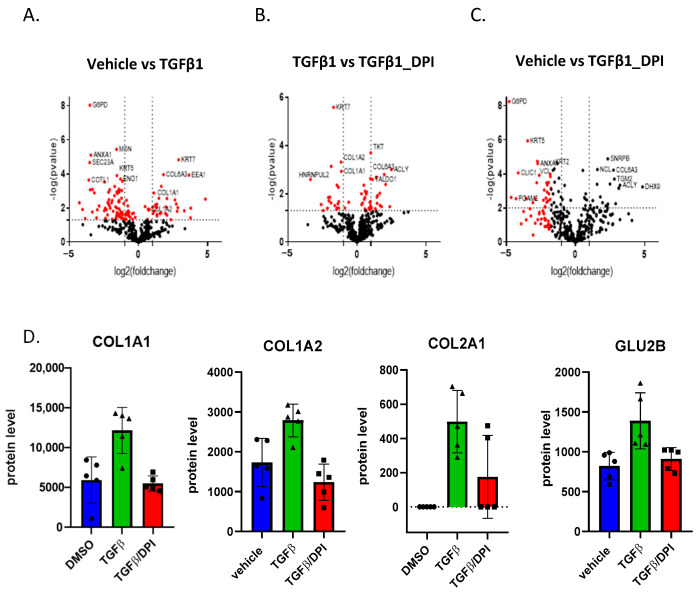
Volcano plots of protein expression changes and proteomic analysis of fibroblasts following TGFβ1 treatment in HFF: (**A**) the higher protein expression of several collagen family members after mediation by TGF1β; (**B**,**C**) the reduction in collagen levels after DPI treatment compared to TGFβ1-treated and vehicle controls. Differentially expressed proteins were identified based on a *p*-value threshold of <0.05 (Mann–Whitney test) and a fold change ratio ≥1.5 or ≤0.66. (**D**) Proteins involved in collagen synthesis (COL1A1, COL1A2, COL2A1, GLU2B). Error bars represent the standard deviation of biological replicates.

**Figure 3 antioxidants-14-00486-f003:**
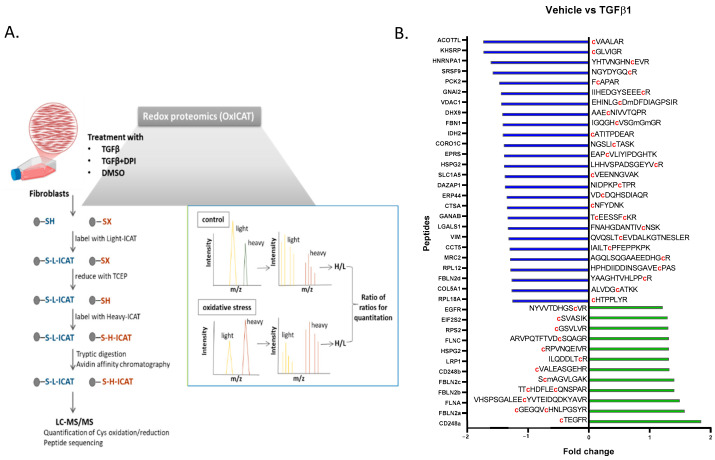
(**A**) Scheme of the method detecting the protein cysteine-residue redox state in fibroblasts treated with TGFβ1, vehicle, or TGFβ1/DPI for 24 h. Cells were incubated with a dissolved cleavable light ICAT reagent for 2 h. After a 30 min incubation with TCEP, reduced cells were mixed with a dissolved cleavable heavy ICAT reagent and incubated for another 2 h. Following overnight tryptic digestion, the biotin tags were cleaved off before separation by liquid chromatography and analysis by mass spectrometry, enabling the simultaneous determination of peptide sequences and the ratios of heavy- and light-labelled cysteine-containing peptides. (**B**) A list of identified peptides in OxiCAT following the TGFβ1 treatment of fibroblasts. Data are expressed as the fold change compared to vehicle-treated cells.

**Figure 4 antioxidants-14-00486-f004:**
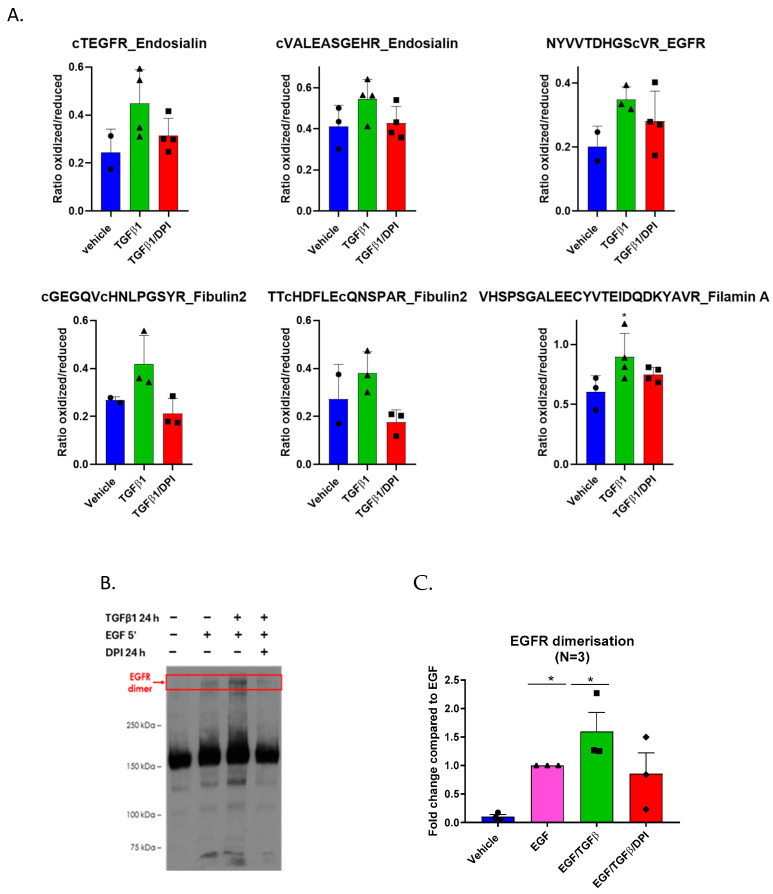
TGFβ1 induces cysteine oxidation and promotes EGFR dimerization. (**A**) Bar graphs showing the ratio of oxidized to reduced cysteine-containing peptides in fibroblasts treated with TGFβ1, a vehicle, or TGFβ1/DPI for 24 h. TGFβ1 treatment increased oxidation levels compared to TGFβ1/DPI. Data represent the mean ± SEM of at least three biological replicates; (**B**) TGFβ1 promotes the dimerization of the epidermal growth factor receptor (EGFR). Western blotting of fibroblasts treated with TGFβ1 and TGFβ1/DPI for 24 h; (**C**) quantification of the EGFR dimers indicated increased stabilization of the EGFR dimer in TGF β1-treated cells. Dimers were lower in DPI-treated cells. Error bars represent the standard deviation of biological replicates. Statistical test and two-way ANOVA, * *p* < 0.05.

**Figure 5 antioxidants-14-00486-f005:**
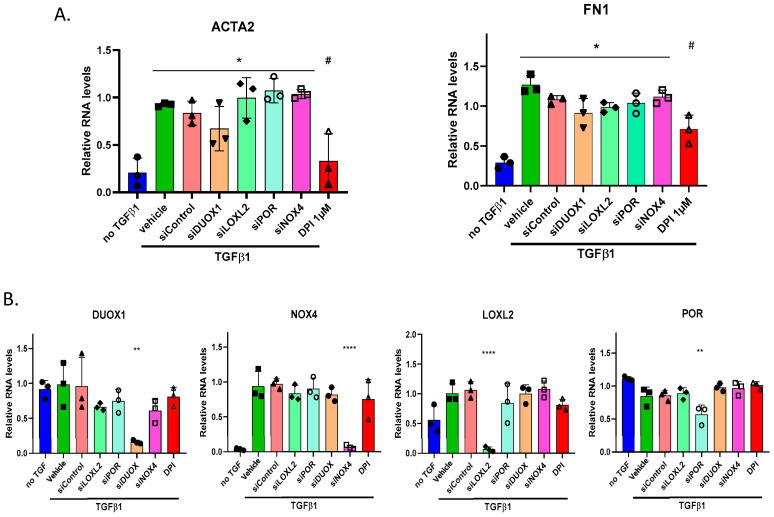
Modulation of myofibroblast marker RNA expression by siRNA: (**A**) qPCR analysis of α-smooth muscle actin (α-SMA) and fibronectin 1 (FN1) RNA expression in HFF cells treated with various siRNA molecules targeting DUOX1, POR, LOXL2, and NOX4, using 1 μM DPI as a control. Cells were exposed to 2 ng/mL TGF-β for 24 h; (**B**) expression levels of target genes following siRNA treatment and TGF-β induction, measured by qPCR after 24 h. Error bars represent the standard deviation of biological replicates. Statistical test and one-way ANOVA with multiple comparison. * *p* < 0.05, ** *p* < 0.01, **** *p* < 0.001, # *p* < 0.05.

**Figure 6 antioxidants-14-00486-f006:**
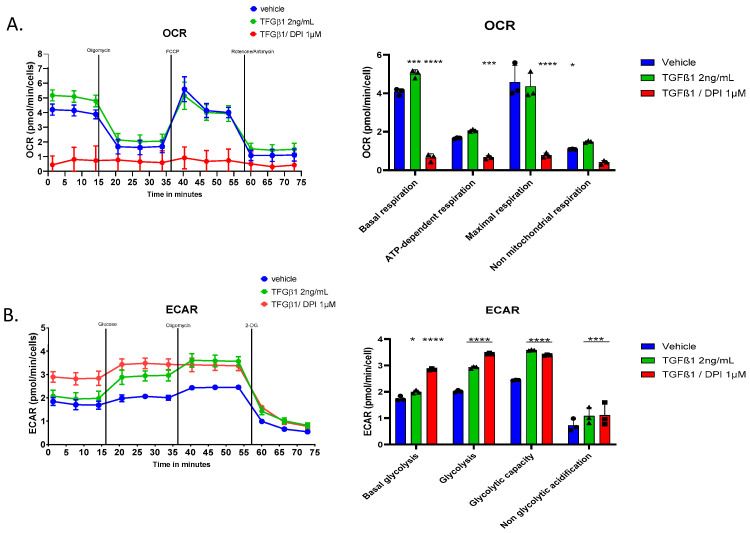
Effects of TGFβ1 and DPI on cellular respiration and glycolysis in HFF. (**A**) Cells were exposed sequentially to oligomycin, FCCP, and rotenone plus antimycin; (**B**) The extracellular acidification rate (ECAR) was measured under the same conditions, with sequential additions of glucose, oligomycin, and 2-DG to evaluate glycolysis. Statistical test and two-way ANOVA, *n* = 3, mean ± SD. * *p* < 0.05, *** *p* < 0.005, **** *p* < 0.001.

## Data Availability

All individual data used to generate the figures and tables are available in the Appendix A.

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
