# Peer review of "Redox Mechanisms Driving Skin Fibroblast-to-Myofibroblast Differentiation"

_antioxidants, 2025, doi:10.3390/antiox14040486_

Round 1

Reviewer 1 Report

Aminzadehanboohi et al, study the detailed interplay between the TGF-B and REDOX pathways in dermal fibroblasts. Through proteomic analysis they identified potential sources of the oxidant generating genes downstream of TGF-B. Through pharmacological inhibition they show the importance of the REDOX pathway in TGF-B mediated myofibroblasts activation. Interestingly they show oxidative processes are important for TGF-B mediated EGFR dimerisation. 

The conclusions drawn from the results are appropriate.

I have a few comments below.

In Figure 1D could the authors expand the panel of pro-fibrotic genes analysed to include FN/PXDN as in figure 1a

In figure 3 A table of selected relevant hits with fold changes would be informative

In figure 4 it would be interesting to observe EGFR localisation in the fibroblasts stimulated with TGF-B/EGF/DPI

N/A

Author Response

Reviewer 1

Comments for Authors

Advice for completing your review can be found at: https://www.mdpi.com/reviewers#Review_Report

Major comments

Aminzadehanboohi et al, study the detailed interplay between the TGF-B and REDOX pathways in dermal fibroblasts. Through proteomic analysis they identified potential sources of the oxidant generating genes downstream of TGF-B. Through pharmacological inhibition they show the importance of the REDOX pathway in TGF-B mediated myofibroblasts activation. Interestingly they show oxidative processes are important for TGF-B mediated EGFR dimerisation. 

The conclusions drawn from the results are appropriate.

We thank reviewer 1 for the global positive evaluation of our manuscript.

I have a few comments below.

In Figure 1D could the authors expand the panel of pro-fibrotic genes analysed to include FN/PXDN as in figure 1a

We selected ACTA2 as a representative and well-established marker of myofibroblast differentiation, as its upregulation is a hallmark of TGFβ1-induced activation. Investigating additional myofibroblast markers would indeed support our findings. We have not tested all antioxidant small molecules on FN1 and PXDN. However, Figure 5B confirms mitigation by DPI of TGFβ1-induced upregulation of FN1.

In figure 3 A table of selected relevant hits with fold changes would be informative

We thank the reviewer for this comment. Fig.3 was updated according to his comment by adding a graph describing fold change between no TGFβ1 vs TGFβ1. The related figure legend and description in results were updated accordingly.

In figure 4 it would be interesting to observe EGFR localisation in the fibroblasts stimulated with TGF-B/EGF/DPI

This is a very good point. A sentence addressing this aspect was added i9n the discussion: “This study is primarily descriptive and does not explore the functional consequences of cysteine redox modulation. In particular, EGFR oxidation following TGF-β1 stimulation may result from changes in its localization, exposing it to different redox environments—an oxidative extracellular space or a reducing intracellular compartment during internalization.”

Reviewer 2 Report

This manuscript explores the redox mechanisms involved in the transformation of myofibroblasts due to TGF, utilizing proteomics approaches. Although the involvement of oxidative stress in TGF-mediated fibrosis is well-established, the methodology applied in this study is innovative. A significant limitation of the research is the absence of evidence supporting NOX4 as a key factor in myofibroblast transformation within this system, leading the authors to speculate that mitochondrial-mediated oxidative stress might play a role. DPI, which was used in the study, can affect multiple enzymes, including xanthine oxidase and nitric oxide synthase, as well as NADPH oxidase. A more detailed investigation into the mechanisms and sources of oxidative stress could enhance the study. Notably, the novel findings from the OxICAT method stand out. The authors should conduct additional experiments to ascertain whether the observed protein oxidation contributes functionally to fibrosis, or if it merely results from increased oxidative stress.

For methods section 2.5.1, does "sample" refer to cell lysate? The steps outlined in this section are somewhat unclear.

In results section 3.3, the description of the pathway analysis results is not very clear. Rather than listing the genes involved, it would be more effective to provide a list of enriched pathways from their cytoscape analysis.

In Figure 2, the labels for panels A and C should be consistent—either both should be labeled as "vehicle" or "DMSO."

In results section 3.7, the knockdown efficiency for POR did not achieve statistical significance. Have the authors considered extending the time points beyond just 24 hours? Given that DPI's effects are non-specific and the authors suggest that mitochondrial-sourced superoxide may be involved, they should consider measuring mitochondrial-specific superoxide with and without DPI. In addition, using a mitochondrial-specific antioxidant such as mitoQ could strengthen their claim.

In results section 3.8, the effect of DPI on ECAR appears counterintuitive. Do the authors have an explanation for how increased glycolysis due to DPI might relate to the pro-fibrotic effects of TGF?

In the conclusions, the authors claim mitochondrial involvement in the process, but this was not demonstrated. It may be premature to conclude this without supporting data.

Author Response

Reviewer 2

Comments for Authors

Advice for completing your review can be found at: https://www.mdpi.com/reviewers#Review_Report

Major comments

This manuscript explores the redox mechanisms involved in the transformation of myofibroblasts due to TGF, utilizing proteomics approaches. Although the involvement of oxidative stress in TGF-mediated fibrosis is well-established, the methodology applied in this study is innovative. A significant limitation of the research is the absence of evidence supporting NOX4 as a key factor in myofibroblast transformation within this system, leading the authors to speculate that mitochondrial-mediated oxidative stress might play a role. DPI, which was used in the study, can affect multiple enzymes, including xanthine oxidase and nitric oxide synthase, as well as NADPH oxidase. A more detailed investigation into the mechanisms and sources of oxidative stress could enhance the study. Notably, the novel findings from the OxICAT method stand out. The authors should conduct additional experiments to ascertain whether the observed protein oxidation contributes functionally to fibrosis, or if it merely results from increased oxidative stress.

We thank reviewer 2 for the global positive evaluation of our manuscript.

Detail comments

For methods section 2.5.1, does "sample" refer to cell lysate? The steps outlined in this section are somewhat unclear.

We are sorry for the inconvenience. Yes, the “sample” refers to the cell lysate. We have revised the text to read better.

In results section 3.3, the description of the pathway analysis results is not very clear. Rather than listing the genes involved, it would be more effective to provide a list of enriched pathways from their cytoscape analysis.

We thank the reviewer for this insightful comment. To make the description of the pathways clearer, a new table has been added -Supplementary Table 4, listing all relevant information, including the list of enriched pathways from the cytoscape analysis and proteins involved in each case.

In Figure 2, the labels for panels A and C should be consistent—either both should be labeled as "vehicle" or "DMSO."

The term vehicle was selected and replaced all over the figures.

In results section 3.7, the knockdown efficiency for POR did not achieve statistical significance.

We verified the data and added a * for significance.

Have the authors considered extending the time points beyond just 24 hours?

We have tested longer treatment of NOX4 siRNA on human fibroblasts and measured no mitigation of ACTA2 expression following TGFβ1 treatment (see figure below).

Given that DPI's effects are non-specific and the authors suggest that mitochondrial-sourced superoxide may be involved, they should consider measuring mitochondrial-specific superoxide with and without DPI. In addition, using a mitochondrial-specific antioxidant such as mitoQ could strengthen their claim.

We agree with the reviewer that data using MitoQ and MitoSOX could further strengthen the claim that mitochondria are involved in redox-mediated fibroblast differentiation. To address this, we have revised the text to soften this claim and included a sentence reflecting this adjustment: “While our findings suggest a role for mitochondrial redox signaling in fibroblast differentiation, further studies using specific mitochondrial-targeted probes, such as MitoQ and MitoSOX, would be needed to directly confirm this involvement”

In results section 3.8, the effect of DPI on ECAR appears counterintuitive. Do the authors have an explanation for how increased glycolysis due to DPI might relate to the pro-fibrotic effects of TGF?

We politely disagree with the reviewer. In fact, inhibition of oxygen mitochondrial respiration most likely a compensatory metabolic shift by increasing glycolysis.

In the conclusions, the authors claim mitochondrial involvement in the process, but this was not demonstrated. It may be premature to conclude this without supporting data.

As stated above, our study is primarily descriptive. While several studies suggest a role for mitochondrial ROS in fibroblast-to-myofibroblast differentiation, we have deliberately avoided making strong claims about this relationship. Nevertheless, we have made several modifications to further soften our statement:

Lane 26: Low concentrations of diphenyleneiodonium mitigated myofibroblast differentiation, mitochondrial oxygen consumption suggesting the involvement of a flavoenzyme.

Lane 99: Finally, our findings suggested a role for mitochondrial activity in fibroblast-to-myofibroblast differentiation.

Lane 728: While our findings suggest a role for mitochondrial redox signaling in fibroblast differentiation, further studies using specific mitochondrial-targeted probes, such as MitoQ and MitoSOX, would be needed to directly confirm this involvement.

Reviewer 3 Report

The paper provides an interesting insight into the mechanism of skin fibroblast to myofibroblasts differentiation involving redox biology hypothesis. This is a very intresting and well-conducted study, providing several very interesing insights. 

I do not have any major comment to make.

Please provide Passage number for the human foreskin fibroblasts used for each experiment, as these are already 16 populations old when purchased, with only 44 populations doubling remaining before potential senescence. 

Also, please comment on impact of growing these cells in an essentially would-healing context of 10% serum. 

1) In the paper there is no information provided about the quality score of RNA used for qPCR (A260/280). The range for all the samples should be specified

2) There is also no description about SYBR Green assay in the paper (which type of the master mix, which provider company)?

3) Provide dilutions of primer used for qPCR. If there is a comparison of the expression between different genes the conditions of performing qPCR should be the same (e.g. similar primer dilutions, the same reagents and protocols on the 7900HTSDS system, Applied Biosystems machine).

4) How much cDNA was used for qPCR reaction. Each primer efficiency factor could be also evaluated in laboratory during dilution curve preparation for each primer and should be provided in the article in the table with sequences of primers (there is a possibility to also evaluate the primers efficiency based on the primer sequences in silico using online available software).

5) There was also no clear information on how the raw data analysis was done after qPCR (by delta delta CT method)? Which software was used for normalization of the expression in relation to housekeeping genes and for statistical data analysis in relation to non-stimulated TGFB1 genes?

6) In relation to detection of H2O2 in fibroblasts, why were the cells seeded on the 96-well plate in 100 ul of Hank’s balanced salt solution (HBSS)? Why for 6 hours before experiment the cell culture medium (still HBSS) was partially removed and the H2O2 estimation assay (Promega) was performed. So, I am assuming the cells were kept in HBSS for more than 6 hours - impact on cell viability? The viability of the cells during the ROS-Glo H2O2 assay should be checked?

7) The Cytoscape platform was used with ClueGO plugin was used. However, this plugin does not indicate the direction of changes; only ranked pathway overrepresented in the tested dataset with statistical significance. Is this any issue? Could the ReactomeFIViz plugin: (https://reactome.org/tools/reactome-fiviz) on Cytoscape be prefered i.e. show the direction of changes (up or down-regulation) of each Reactome pathway with statistical significance after overlapping the molecule expression scores into the pathways. Ingenuity pathway analysis, IPA (Qiagen) software could be also used for this purpose.  

8) In relation to gene silencing in HFF description by siRNA the incubation of the cells for 24h in transfection mixture with siRNA was short. Usually is recommended to incubate the cells with siRNA constructs and additional reagents for 2-4 days to see effective changes on gene/protein expression. Especially for protein expression changes evaluation after gene silencing with siRNA the incubation time should be increased up to 4 days. Also, there is no specification how the siRNA incubation was ended and how the cells were collected for further analysis.

Author Response

Reviewer 3

Major comments

The paper provides an interesting insight into the mechanism of skin fibroblast to myofibroblasts differentiation involving redox biology hypothesis. This is a very intresting and well-conducted study, providing several very interesing insights. 

I do not have any major comment to make.

We thank reviewer 3 for the careful evaluation of the methodological approach of our study.

Please provide Passage number for the human foreskin fibroblasts used for each experiment, as these are already 16 populations old when purchased, with only 44 populations doubling remaining before potential senescence. 

HFF were passaged maximum 7 times. This has been added in Material and method section, lane 118.

Also, please comment on impact of growing these cells in an essentially wound-healing context of 10% serum. 

HFF were kept in a serum-free medium for 24 hours before addition of the pro-fibrotic factor TGFβ1.

Detail comments

1) In the paper there is no information provided about the quality score of RNA used for qPCR (A260/280). The range for all the samples should be specified

The RNAeasy Mini Kit from Qiagen typically generate RNA of high purity. The A260/A280 ratio is always in the range of 1.9-2.1. The following sentence was added in the method section: Concentration and purity (A260/A280 in the range of 1.9-2.1) were evaluated using a Nanodrop Spectrometer (Thermo Fisher Scientific).

2) There is also no description about SYBR Green assay in the paper (which type of the master mix, which provider company)?

We used the PowerUp SYBR Green Master Mix from Applied Biosystems by Thermo Fisher Scientific Ref : A25743. This now added in the methods.

 3) Provide dilutions of primer used for qPCR. If there is a comparison of the expression between different genes the conditions of performing qPCR should be the same (e.g. similar primer dilutions, the same reagents and protocols on the 7900HTSDS system, Applied Biosystems machine).

Final concentration of primers is 0.3 µM. In qPCR experiments, as each primer set has specific characteristics, we do not compare between genes, but between conditions for a single gene.

4) How much cDNA was used for qPCR reaction. Each primer efficiency factor could be also evaluated in laboratory during dilution curve preparation for each primer and should be provided in the article in the table with sequences of primers (there is a possibility to also evaluate the primers efficiency based on the primer sequences in silico using online available software).

We used 10 ng cDNA

5) There was also no clear information on how the raw data analysis was done after qPCR (by delta delta CT method)? Which software was used for normalization of the expression in relation to housekeeping genes and for statistical data analysis in relation to non-stimulated TGFB1 genes?

Normalisation was used as follows: Relative quantities (RQ) were calculated with the formula RQ=E–Ct using efficiencies (E) calculated for each run with the DART-PCR algorithm, as described (Peirson et al., 2003). A mean quantity was calculated from triplicate PCR reactions for each sample, and this quantity was normalized to two or three similarly measured quantities of normalization genes as described (Vandesompele et al., 2002). Normalized quantities were averaged for three replicates for each data point and represented as the mean±s.d. The highest normalized relative quantity was arbitrarily designated as a value of 1.0. Fold changes were calculated from the quotient of means of these normalized quantities and reported as ±s.d. The statistical significance of fold-changes was determined by a paired Student’s t-test.

6) In relation to detection of H2O2 in fibroblasts, why were the cells seeded on the 96-well plate in 100 ul of Hank’s balanced salt solution (HBSS)? Why for 6 hours before experiment the cell culture medium (still HBSS) was partially removed and the H2O2 estimation assay (Promega) was performed. So, I am assuming the cells were kept in HBSS for more than 6 hours - impact on cell viability? The viability of the cells during the ROS-Glo H2O2 assay should be checked?

Measuring Hâ‚‚Oâ‚‚ in HFFs is highly challenging. We tested several methods to assess Hâ‚‚Oâ‚‚ kinetics, including Amplex Red/HRP, L-012, and coumarin boronic acid, and maintained cells in HBSS for 4–6 hours without observing any viability issues. Among these approaches, only RosGlo detected a statistically significant difference between fibroblasts and myofibroblasts. We believe this is because RosGlo measures both intracellular and extracellular Hâ‚‚Oâ‚‚, and the observed difference is unlikely to be due to changes in HFF viability.

7) The Cytoscape platform was used with ClueGO plugin was used. However, this plugin does not indicate the direction of changes; only ranked pathway overrepresented in the tested dataset with statistical significance. Is this any issue? Could the ReactomeFIViz plugin: (https://reactome.org/tools/reactome-fiviz) on Cytoscape be prefered i.e. show the direction of changes (up or down-regulation) of each Reactome pathway with statistical significance after overlapping the molecule expression scores into the pathways. Ingenuity pathway analysis, IPA (Qiagen) software could be also used for this purpose.  

We thank the reviewer for this well taken comment. We performed the Cytoscape analysis including the fold change, (reflecting if the protein was up- or down- regulated) in the input data. As a result, the output provides the information regarding the percentage of up or down- regulated proteins in each pathway, as an indication of the respective pathway activity –(under column ‘specificity ‘ ; -Supplementary Table 4).  We realize that this information can only be suggestive (and not conclusive) with respect to the pathway activity, yet it is still informative and can guide future studies. We agree Ingenuity would be more appropriate for such analysis, however this is a commercially available software and we do not currently have the respective license. A new table, Supplementary table 4 has been added to provide the detailed results from the pathway analysis.

8) In relation to gene silencing in HFF description by siRNA the incubation of the cells for 24h in transfection mixture with siRNA was short. Usually is recommended to incubate the cells with siRNA constructs and additional reagents for 2-4 days to see effective changes on gene/protein expression. Especially for protein expression changes evaluation after gene silencing with siRNA the incubation time should be increased up to 4 days. Also, there is no specification how the siRNA incubation was ended and how the cells were collected for further analysis.

We tested the siRNA targeting NOX4 for up to 78 hours without observing mitigation of expression of myofibroblast markers (see graph of the answer nr 2). The incubation was ended by removing the medium, washing with PBS 1X and directly adding the RLT solution from the lysing the cells using the RLT solution of RNAeasy Mini Kit from Qiagen.

Round 2

Reviewer 1 Report

The authors have addressed my questions with additional data and information.

N/A

Author Response

We thank the reviewer

Reviewer 2 Report

The authors did not provide a satisfactory response to the question:

In the results section 3.8, the effect of DPI on ECAR seems counterintuitive. Could the authors explain how increased glycolysis due to DPI might relate to the pro-fibrotic effects of TGF?

Perhaps my question was unclear. It is well-established through numerous publications that TGF induces glycolysis, which in turn promotes fibrosis. Therefore, what does the increase in glycolysis in the presence of DPI suggest in this context? Could the authors provide a more detailed explanation instead of a brief answer dismissing this inquiry?

The authors did not provide a satisfactory response to the question:

In the results section 3.8, the effect of DPI on ECAR seems counterintuitive. Could the authors explain how increased glycolysis due to DPI might relate to the pro-fibrotic effects of TGF?

Perhaps my question was unclear. It is well-established through numerous publications that TGF induces glycolysis, which in turn promotes fibrosis. Therefore, what does the increase in glycolysis in the presence of DPI suggest in this context? Could the authors provide a more detailed explanation instead of a brief answer dismissing this inquiry?

Author Response

We apologize for not adequately addressing the question in our previous response. We did not fully understand the underlying nature of the inquiry. Below, we provide a more comprehensive explanation and hypotheses to reconcile the apparent contradiction between the antifibrotic effects of DPI and its role in promoting glycolysis.

Although both DPI and TGFβ1 enhance glycolysis, the associated metabolic and signaling contexts differ markedly. TGFβ1-induced glycolysis occurs in the context of a fully active mitochondrial metabolism. This supports robust biosynthetic activity and ATP production, both of which are essential for myofibroblast differentiation. DPI, in contrast, strongly inhibits mitochondrial function, thereby forcing cells to rely on aerobic glycolysis for ATP generation. This creates a profoundly different metabolic environment, likely incompatible with the biosynthetic processes required for myofibroblast differentiation. DPI-driven glycolysis under mitochondrial inhibition leads to metabolic stress, impaired anabolic capacity, and suppression of ROS signalling—ultimately preventing myofibroblast differentiation. Altogether, these observations suggest that while increased glycolysis is associated with myofibroblast differentiation, it is not necessarily causative. Instead, other pathways (which are inhibited by DPI), such as mitochondrial metabolism and ROS signalling, appear to play more critical roles.

We thank gain the reviewer for this question and we hope that we provided a satisfactory answer. To address this point, we added the following sentence in the manuscript (lane 707):

“Both TGFβ1 and DPI enhanced glycolysis. However, DPI simultaneously impaired mitochondrial function and increased ROS production, creating a metabolic environment that may be incompatible with myofibroblast differentiation and highlighting the fact that glycolysis alone may not be sufficient to drive fibroblast to myofibroblast differentiation”.